# Skewness-Based Partitioning in SpatialHadoop

**Alberto Belussi [1,†]**, **Sara Migliorini [1,*,†]** **and Ahmed Eldawy [2,†]**

1. Department of Computer Science, University of Verona, 37134 Verona, Italy; alberto.belussi@univr.it
2. Department of Computer Science and Engineering, University of California, Riverside, CA 92521, USA; eldawy@ucr.edu
* Correspondence: sara.migliorini@univr.it
† These authors contributed equally to this work.

**Abstract:** In recent years, several extensions of the Hadoop system have been proposed for dealing with spatial data. SpatialHadoop belongs to this group of projects and includes some MapReduce implementations of spatial operators, like range queries and spatial join. the MapReduce paradigm is based on the fundamental principle that a task can be parallelized by partitioning data into chunks and performing the same operation on them, (map phase), eventually combining the partial results at the end (reduce phase). Thus, the applied partitioning technique can tremendously affect the performance of a parallel execution, since it is the key point for obtaining balanced map tasks and exploiting the parallelism as much as possible. When uniformly distributed datasets are considered, this goal can be easily obtained by using a regular grid covering the whole reference space for partitioning the geometries of the input dataset; conversely, with skewed distributed datasets, this might not be the right choice and other techniques have to be applied. for instance, SpatialHadoop can produce a global index also by means of a Quadtree-based grid or an Rtree-based grid, which in turn are more expensive index structures to build. This paper proposes a technique based on both a box counting function and a heuristic, rooted on theoretical properties and experimental observations, for detecting the degree of skewness of an input spatial dataset and then deciding which partitioning technique to apply in order to improve as much as possible the performance of subsequent operations. Experiments on both synthetic and real datasets are presented to confirm the effectiveness of the proposed approach.

**Keywords:** SpatialHadoop; skewed data; partitioning; MapReduce; BigData

## 1. Introduction

In recent years several application contexts require the analysis of huge amount of data and very frequently the dimensions of interest include spatial properties. Therefore, many efforts have been devoted by researchers to the implementation of solutions for efficiently performing such kind of computations. The MapReduce paradigm has also been successfully applied to implement parallel solution for those spatial operations that are typically required for performing spatial data analysis. In particular, the well-known range query, spatial join and $k$-nearest neighbor operations are currently available in many MapReduce frameworks. For instance, SpatialHadoop [1], a spatial extension of Apache Hadoop [2], provides all these operations, in some cases also in different variants, which can be combined with different partitioning techniques, usually called global indexing strategies.

The fundamental principle of the MapReduce paradigm is the subdivision of the input into independent chunks (dataset partitioning) on which the same operation can be performed in parallel

(map phase), possibly combining the partial results at the end in a successive task (reduce phase). Therefore, one of the main aspects that can have a great impact on the effectiveness of the parallel execution is the partitioning of the input dataset. A good partitioning strategy has to produce uniform chunks in order to ensure balanced map tasks, that is, tasks whose execution should require more or less the same amount of resources to be processed.

Many MapReduce frameworks, including Hadoop, natively supports a default partitioning method based on data size, namely in this case the goal is to produce chunks, called *splits*, having almost the same size in bytes. This aseptic technique can be effective for traditional (textual) data processing, but may not be the best choice for partitioning spatial data. Indeed, it could produce poor performances in those cases in which the input data have to be pruned or filtered considering its spatial location. Conversely, more appropriate partitioning techniques could separate spatially nearby records in different partitions [3], favouring the subsequent data analysis.

For this reason, several spatial partitioning methods have been implemented in specific extensions of MapReduce frameworks in order to subdivide the geometries into splits according to their spatial properties. For instance, indexes based on a regular grid, Quad-trees and R-trees are available in SpatialHadoop [4] and can be applied to partition a dataset before executing a given operation. However, in this paper we show that *not all spatial partitioning techniques behave in the same way* and in different cases *the best technique to use can change* according to the *spatial characteristics of the datasets* at hand and eventually the operation that has to be performed on the partitioned datasets.

As a first example of the kind of issue we want to consider in this paper, we shown in Table 1 the results of the execution in SpatialHadoop of the Distributed Join (DJ) [5], the Range Query (RQ) and of the *k*-Nearest Neighbor operation (*k*-NN) when applied to different situations. We consider the following cases:

1. the DJ operation applied to two datasets which are both uniformly distributed. In particular, the first one (denoted as $D1_{UD}$) is partitioned using a regular grid (*GR*), while the second one (denoted as $D2_{UD}$) has been partitioned using different techniques, namely regular grid (*GR*), Quadtree (*QT*) and R-tree (*RT*).
2. the DJ operation applied to a uniformly distributed dataset (denoted as $D1_{UN}$) and a skewed one (denoted $D2_{SKW}$). In particular, $D1_{UN}$ has been partitioned using a regular grid, while $D2_{UN}$ has been partitioned with several possible partitioning techniques.
3. the RQ operation applied to a skewed dataset ($D1_{SKW}$) partitioned with several possible partitioning techniques.
4. the *k*-NN operation applied to a skewed dataset ($D1_{SKW}$) partitioned with several possible partitioning techniques.

In Table 1, the second column reports the applied partitioning technique, the third column contains the total time in milliseconds for performing the operation, while the last column reports some statistics about the map tasks. In particular, we indicate the number of instantiated map tasks, the average time taken by them and the relative standard deviation (RSD) between their execution times. A greater value of RSD means that there is a great difference between the execution time of different map tasks, while a lower value means that they essentially take the same time to complete.

As expected, when both datasets are uniformly distributed, the response time of the DJ is similar regardless of the used index, while, when a skewed distributed dataset is considered, then the differences are significant and in this particular case are in favor of the R-tree. This is mainly due to the fact that when the distribution is skewed, the partitioning of the geometries based on a regular grid does not produce balanced splits (i.e., RSD is about 90%), while the Quadtree and the R-tree indexes perform better and produce more balanced splits (i.e., RSD is about 28% for the R-tree). A similar behaviour

can be observed for the other two operations, RQ and KNN, where again the Quadtree and R-tree partitioning techniques perform better. In particular, also in this case the best technique is the one with the lowest deviation between the execution time of the map tasks. Therefore, we can observe that balancing the cost of the map tasks is crucial for effectively exploiting the parallel potentiality provided by a MapReduce framework.

**Table 1.** Execution of the DJ, RQ and KNN operations in SpatialHadoop with datasets having different kind of partitioning (i.e., partitioning based on: GR = regular grid, QT = Quadtree or RT = R-tree) and having different distributions (i.e., UN = uniform distribution, SKW = skewed distribution). RSD is the standard deviation of running times between map tasks.

| Dataset and Operation | Partitioning Technique | Tot. Time (Millis) | # Tasks | Map Tasks AVG Time (Millis) | %RSD Time |
|---|---|---|---|---|---|
| $D1_{UN} \bowtie D2_{UN}$ | GR/GR | 145,307 | 37 | 15,833 | 4% |
| $D1_{UN} \bowtie D2_{UN}$ | GR/QT | 150,458 | 51 | 18,902 | 9% |
| $D1_{UN} \bowtie D2_{UN}$ | GR/RT | 147,646 | 54 | 16,231 | 7% |
| $D1_{UN} \bowtie D2_{SKW}$ | GR/GR | 125,327 | 33 | 22,710 | 90% |
| $D1_{UN} \bowtie D2_{SKW}$ | GR/QT | 96,001 | 52 | 11,209 | 50% |
| $D1_{UN} \bowtie D2_{SKW}$ | GR/RT | **40,205** | 21 | 18,087 | **28%** |
| $Range(D1_{SKW})$ | GR | 37,798 | 3 | 14,647 | 29% |
| $Range(D1_{SKW})$ | QT | 37,390 | 6 | 7353 | 67% |
| $Range(D1_{SKW})$ | RT | **32,793** | 3 | 13,096 | **9%** |
| $Knn(D1_{SKW})$ | GR | 47,861 | 3 | 14523 | 21% |
| $Knn(D1_{SKW})$ | QT | 37,639 | 2 | 12,324 | 57% |
| $Knn(D1_{SKW})$ | RT | **27,786** | 2 | 4632 | **13%** |

The aim of this paper is to provide a way to easily detect some hints about the dataset distribution and based on them chose the more effective partitioning technique to apply. In this manner the partitioning will be mainly based on the spatial properties of the geometries contained in the dataset and not only on its size in bytes.

In literature, many statistical techniques have been proposed to provide a summarized description of a dataset. These descriptors, often called *sketches*, are used to speed up the query processing by providing approximate answers based on them [6]. One of their main uses in spatial big data analysis can be the estimation of selectivity for a join operation. we can classify relevant sketching techniques into two main categories—*sampling-based methods* [1,4,7,8] and *histogram-based methods* [9,10]. In general, histogram-based methods are shown to be superior for accurate spatial selectivity estimation [11,12].

This paper proposes a new technique which internally constructs multiple uniform histograms. However, it does not require to store and maintain all of them, since, rather, it aggregates them into a regression model. The regression model produces only two numbers that are used as indicators of the dataset distribution in the reference space. More specifically, the proposed technique is based on the concept of box-counting that was first proposed in Reference [13] for computing the fractal dimension of a dataset of points. The behavior of the box-counting function measured in a restricted range of values (representing the cell side of the grid) can be described by a power law and it was used in Reference [14] for estimating the selectivity of self-join and range query, then extended in Reference [15] to the spatial join on distinct datasets. We propose its application in the context of big spatial data for the following reasons: (i) it is an efficient technique for detecting information about the dataset distribution; (ii) it produces just one number characterizing the data distribution and does not require to store auxiliary structures, like histograms; (iii) the box-counting function can be computed in parallel, since it calculates a uniform histogram storing the counts and this can be easily implemented in MapReduce.

The main contributions of the paper are: (i) the extension of the box-counting function to all types of geometries; previously, it could be applied only to a set of points; (ii) the definition of a MapReduce algorithm for its efficient computation, possibly on a sample of the considered dataset; (iii) the definition of heuristics for the choice of the best partitioning technique for a given datasets whose distribution has been described by means of the box-counting function. A subset of these results have already been presented in Reference [16]. In particular, this paper considerably extends the work done in Reference [16] by providing:

(a) an analytical presentation of the box-counting function together with some examples that justify the proposed approach (Section 3),

(b) a detailed description of a MapReduce implementation of the algorithm for computing the exponents of the power law that characterizes the box-counting function (Section 4),

(c) an extended presentation of the proposed heuristic for choosing the right partitioning techniques together with the proofs of the defined properties (Section 5),

(d) a comprehensive set of experiments performed both on real and synthetic datasets (Section 6) which includes three types of operation: the spatial join, the range query and the *k*-nearest neighbor.

## 2. Background

In this section we summarize the main characteristic of the MapReduce implementation of spatial operations like spatial join, range query and K-nearest neighbor, together with the main partitioning technique usually available in cluster systems dedicated to spatial data, such as SpatialHadoop.

### 2.1. Partitioning Techniques in SpatialHadoop

This section briefly describes the partitioning techniques available in SpatialHadoop and shows their effect on skewed distributed datasets.

At the basis of the MapReduce paradigm is the idea to divide the input dataset into fixed-size blocks, called *splits*, on which the same operation (map task) can be instantiated and executed in parallel. If all map tasks can be executed in parallel, then the total execution time depends on the map task that takes longer. Therefore, the fastest parallel executions can be obtained when the map tasks are well balanced. It follows that the partitioning of data into splits is a crucial operation for obtaining well balanced map tasks and ensure faster executions. Hadoop traditionally applies a random division of the input data, during split generation the only prescribed constraint regards the size in bytes of such splits on the HDFS (Hadoop Distributed File System). However, this naïve partitioning cannot be the right choice during spatial analysis for which some filtering or pruning is always performed for evaluating spatial predicates.

Some partitioning techniques that take into account the spatial correlation (locality property) of the input data is necessary, that is, geometries that are closed in space will be placed in the same split. Table 2 compares different partitioning techniques when applied to different datasets. For each dataset and applied technique, we report the number of produced splits and the relative standard deviation (RSD) between their cardinalities (i.e., degree of balancing in terms of number of geometries). The simplest one relies on the use of a uniform grid (first row). However, such technique might produce balanced tasks only for uniformly distributed datasets, but not in general. Indeed, as we have shown in Table 1, when skewed distributed datasets are considered the cost of the map tasks is often unbalanced, causing a performance degradation (% RSD is 1% for the uniform distribution, while it is more than 100% for the other datasets). Therefore, in order to guarantee the generation of balanced splits, different types of grids should be used for data partitioning. In SpatialHadoop three main types of grid exists for global data partitioning, the first two are space-based while the last one is a data-based partitioning:

- *Regular grid*: it identifies the cells by dividing the 2D space in both axes by a constraint measure; it is suitable only for uniformly distributed datasets;
- *Quadtree-based grid*: it identifies the cells by recursively subdividing a cell (starting from the whole space) in 4 equal cells until the number of geometries per cell reaches a threshold;
- *Rtree-based grid*: it identifies the cells by recursively aggregating the geometries of the dataset until the number of geometries per cell reaches a threshold.

**Table 2.** Global index grids for synthetic and real datasets.

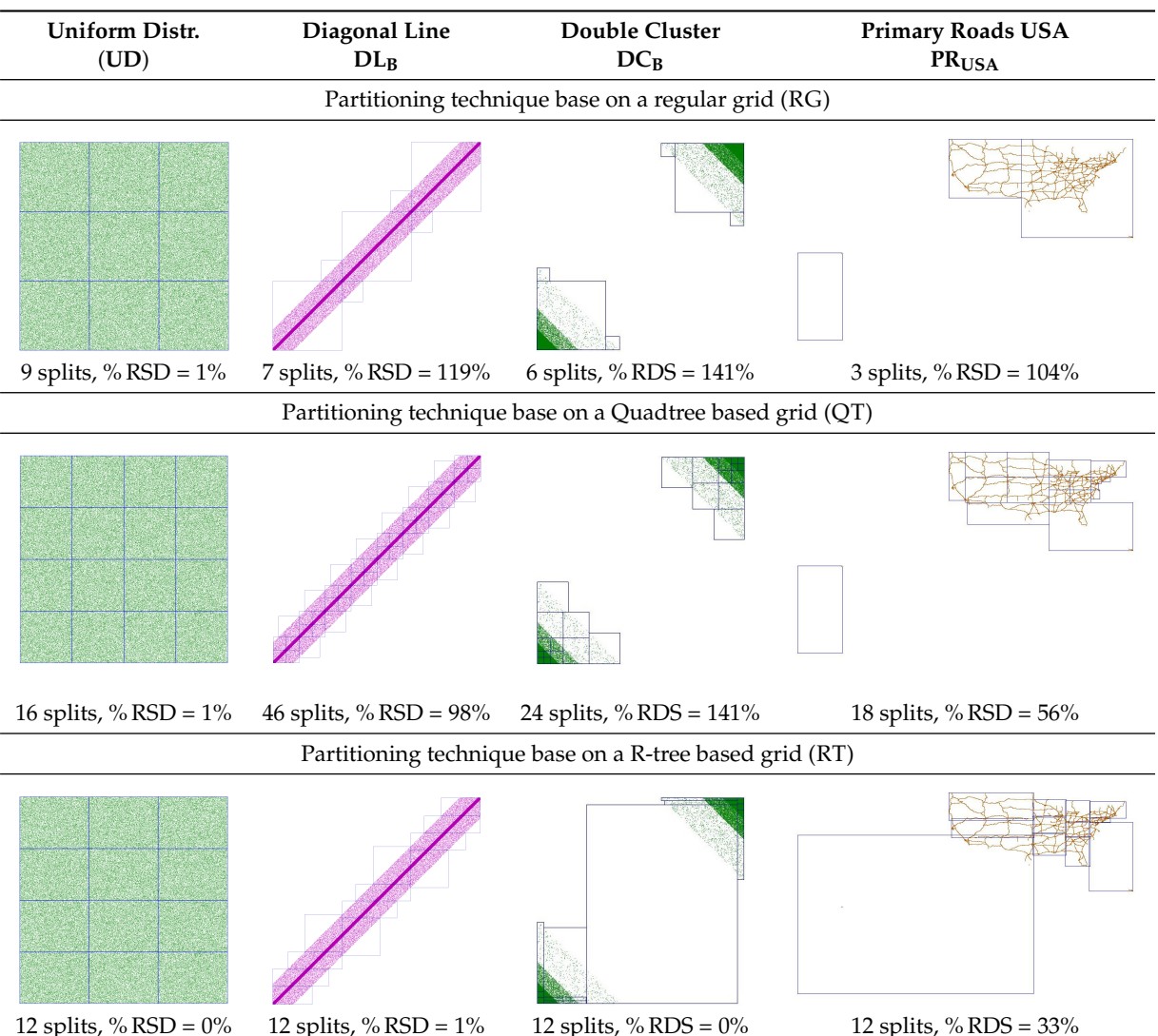

| Uniform Distr. (UD) | Diagonal Line $DL_B$ | Double Cluster $DC_B$ | Primary Roads USA $PR_{USA}$ |
|---|---|---|---|
| | Partitioning technique base on a regular grid (RG) | | |
| 9 splits, % RSD = 1% | 7 splits, % RSD = 119% | 6 splits, % RDS = 141% | 3 splits, % RSD = 104% |
| | Partitioning technique base on a Quadtree based grid (QT) | | |
| 16 splits, % RSD = 1% | 46 splits, % RSD = 98% | 24 splits, % RDS = 141% | 18 splits, % RSD = 56% |
| | Partitioning technique base on a R-tree based grid (RT) | | |
| 12 splits, % RSD = 0% | 12 splits, % RSD = 1% | 12 splits, % RDS = 0% | 12 splits, % RDS = 33% |

We partition synthetic and real datasets using the above listed techniques and we obtain the grids shown in Table 2. As you can notice, the R-tree is the one that ensures the best balancing in terms of number of geometries per cell (i.e., % RSD), but is some cases (e.g., the $DC_B$ or the $PR_{USA}$) it can produce very unbalanced cells in terms of covered space.

## 2.2. Spatial Operations in SpatialHadoop

In order to evaluate the effectiveness of the proposed method in choosing the right partitioning techniques for spatial datasets of any distribution, we consider three operations: spatial join, range query and *k*-nearest neighbor. Hereby we briefly describe their implementation in MapReduce.

- *Distributed Spatial join* (DJ)—The distributed spatial join operator (DJ) works on two input datasets $D_1$ and $D_2$. Both datasets can be partitioned according to one of the index techniques mentioned in the previous section. Therefore, each dataset $D_i$ is represented as a collection of partitions, $D_i = \{s_{i,1}, ..., s_{i,k_i}\}$. Each partition $s_{i,j}$ is characterized by a key, representing the index cell, and a list of geometries belonging to that cell. Given two datasets $D_1$ and $D_2$, the input of each map task is prepared by a binary reader which is able to generate a compound split $c_i = (s_{1,x}, s_{2,y})$ starting from a pair of splits belonging to the two datasets (i.e., $s_{1,x} \in D_1$ and $s_{2,y} \in D_2$). In particular, a filter is used for preparing the compound splits, so that only the pairs of splits regarding intersecting cells are generated. Therefore, the number of generated compound splits, and consequently the number of map tasks, is equal to the number of pairs of intersecting cells. Given a compound split, each map tasks loads its content into two lists and then it applies a Plane-sweep algorithm for checking the intersection between their geometries. DJ is a map-only job, no reduce phase is necessary, and clearly it can be classified as a map-side join.
- *Range query* (**RQ**)—the range query (RQ) works on one file only, which can be partitioned in splits using one of the available techniques, $D = \{s_1, ..., s_k\}$. Since also in this case, the reader works on indexed data, given a query rectangle $R$, a filter is applied in order to select the splits (cells) that intersects $R$. A map task is instantiated for each split (cell) having a non-empty intersection with $R$. Inside each map tasks, a test is performed in order to precisely determine which split geometries intersect $R$. RQ is also a map-only job, namely it has no reducers.
- *k-Nearest Neighbor* (*k*-**NN**)—As the range query also *k*-NN reads only one dataset $D = \{s_1, ..., s_k\}$. This operation is divided in more than one MapReduce job. In particular, given a dataset $D$ and a query point $Q$, the first job tries to find the $k$ nearest neighbor geometries in the split $s_i$ which correspond to the cell $c_i$ containing the query point $Q$. This job is composed of a map and a reduce phase. During the map phase, the distance between each geometry $g \in s_i$ and the point $Q$ is computed. Then, a single reduce task receives the list of such geometries with their distance from $Q$ and extracts the first $k$ points according to that distance. After this job, a circle with center in $Q$ and having as radius the distance of the $k$-th geometry (or the last retrieved one) is computed. If the circle is contained in the cell of $s_i$ the execution terminates; otherwise, the circle is used as filter for the splits to be processed by the second job. More iterations can be done only if the first job does not retrieve $k$ geometries. of course, if $k$ is small very often the first job is sufficient.

## 3. Evaluation of Dataset Skewness

Considering the case study shown in Table 1, it is clear that an easy and efficient way for evaluating the skewness of a spatial dataset can be crucial for choosing the right partitioning technique. This section presents the definition of the box-counting function $BC_D^q(r)$ for a given dataset $D$ containing 2D geometries of type point, line or polygon embedded in the Euclidean plane. This is the first contribution of the paper, since it is an extension of the box-counting function proposed in Reference [14] which originally applies only to finite set of points. The implications of such extension on the estimation correctness are discussed and the possible alternatives are evaluated. We will see later how the analysis of this function can provide some hints about the skewness of the dataset.

**Definition 1** (Box-counting function). *Given a dataset D, containing 2D geometries (i.e, points, lines or polygons), and a scale r, representing the cell size of a grid covering the reference space of D (i.e., the MBR of the whole dataset D), the function Box-counting $BC_D^q(r)$ is defined as follows:*

$$BC_D^q(r) = \sum_i p_i(D)^q \qquad\qquad with\ q \neq 1, \qquad\qquad (1)$$

*where $p_i(D) = count(geometries\ of\ D\ intersecting\ the\ cell\ i)$. The case $q = 1$ is excluded, since it always returns the total number of geometries in the input.*

In Reference [14] the author shows that the box-counting function is useful for computing the generalized fractal dimension of a finite set of points, where $q$ represents the exponent in Equation 1 and $r$ is the considered scale (i.e., the cell side of the grid). Intuitively, given a grid with cells of side $r$, the box-counting function with $q = 0$ proposed in Reference [14] counts the number of cells that are intersected by at least one point of $D$, similarly here the function $BC_D^0(r)$ counts the number of cells that are intersected by at least one geometry of $D$. In this way, when exponents $q$ is greater than 1, the box-counting is the sum of the number of geometries intersecting a cell raised to $q$. As we will show, this function can be used to detect the skewness of a dataset by computing it for $q = 0$ and $q = 2$ while varying the value of $r$. More specifically, the level of skewness of a dataset depends on how this value changes while increasing $r$.

Notice that with respect to the definition given in Reference [14], which applies only to set of points and counts the number of points contained in a cell, in Definition 1 we propose to count the number of geometries that intersect a cell. This extension of the box-counting function from set of points to generic geometries could be obtained in three ways, as discussed below.

(i)　a first option could be to choose a representative point for each geometry of the dataset (e.g., its centroid) and then apply the classical box-counting function. This can be the simplest solution, but it does not ensure to always detect the real behavior of the dataset. This is for instance the case of a datasets containing a set of big polygons covering the whole reference space, if they are approximated by their centroids, the resulting points are all clustered in a small region of the same space, thus producing a point set that does not describe at all the original dataset layout.

(ii)　Another solution could be to substitute the geometries with their vertices; again, there could be regions covered by geometries that are not covered by their vertices, with the same effect described in the previous case.

(iii)　the last solution is the one adopted by this paper, namely to count the number of geometries intersecting a cell, which is equivalent to suppose that each geometry $g$ is converted in a set of points $\mathcal{P}(g)$ covering the same space with a granularity that satisfies the following hypothesis: *if g intersects a cell i, then there exists at least one point $p \in \mathcal{P}(g)$ such that p intersects the cell i.* With this hypothesis, we can extend the box-counting function from point sets to sets of geometries and apply to our case the results of Reference [14]. Notice that even if this solution can produce an over-estimation for the selectivity, it does not have negative effects for the problem considered in this paper, namely the estimation of the dataset distribution, conversely it leads to a more precise result.

**Definition 2.** *Given a dataset D, containing 2D geometries (i.e, points, lines or polygons), the Box-counting plot is the plot of $BC_D^q(r)$ versus r in logarithmic scale. Now, we can consider such plot and exploit the following observation of Reference [14]—for real datasets the box-counting plot reveals a trend of the box-counting function that, in a large interval of scale values r, behaves as a power law:*

$$BC_D^q(r) = \alpha \cdot r^{E_q}, \qquad\qquad\qquad (2)$$

*where $\alpha$ is a constant of proportionality and $E_q$ is a fixed exponent that characterizes the power law.*

As we will see in Example 1, the Box-counting plot is vital for the computation of the exponent $E_q$ for a given dataset $D$, since this exponent becomes the slope of the straight line that approximates $BC_D^q(r)$ in a range of scales $(r_1, r_2)$, thus it can be computed by a linear regression procedure. The exponent $E_q$ characterizes the dataset distribution as explained by the following set of properties.

- for $q = 0$, $E_0$ is negative and the power law, given the length of the cell side $r$, computes the number of cells that are intersected by the dataset $D$. Notice that, if $D$ is uniformly distributed in the reference space (the Euclidean plane in our case), then the number of cells intersecting $D$ coincides with the total number of cells of the grid, thus the more $r$ increases the more this number decreases according to the area of the cells. As a consequence in case of an uniform distribution, $E_0$ is equal to minus the dimension of the embedding space, in our case $E_0 = -2$.
- for datasets that represent fractals (like the Sierpinski's triangle), it is known from the theory that $E_q$ coincides with the fractal dimensions of the fractal for any $q$ (it is a consequence of the self similarity property), thus for the Sierpinski's triangle $E_0 = -1.585$.
- Finally, we can observe that $E_0$ and $E_2$ could be chosen as reference descriptors for a dataset $D$ with the aim to have some hints about the distribution of the geometries in the reference space of $D$. Indeed, $E_0$ can be an indicator of the cases where the dataset leaves empty some areas of the reference space, while $E_2$ can also be affected by the concentration of the datasets in some areas with respect to other ones, that is, the situations where there are no empty areas, but different concentrations in different areas.

**Example 1.** *Two examples of Box-counting plot with $q = 0$ and $q = 2$ are reported in Figure 1, where the considered dataset is shown on the left and the corresponding plots on the right. In particular, the first dataset $D_{sier}$ (Figure 1a) is synthetically produced by a generator of points belonging to the Sierpinski's triangle [17], which is a well known fractal, and by substituting the points with small polygons. The Box-counting plot of $BC_{Sier}^0(r)$ and $BC_{Sier}^2(r)$ are represented by diamonds in Figure 1b,c, respectively. The second dataset $D_{PRaus}$ (Figure 1c) is a real dataset containing a set of lines representing the main roads of Australia; again the Box-counting plot of $BC_{PRaus}^0(r)$ and $BC_{PRaus}^2(r)$ are represented by diamonds in Figure 1e,f, respectively. The Box-counting plots reports also, inside blue and red rectangles, the exponent $E_0$ (Figure 1b,e) and $E_2$ (Figure 1c,f), computed by applying the MapReduce algorithm illustrated in the next section.*

*For the dataset $D_{Sier}$, two intervals of $r$ values with constant slope are detected in the plot $BC_{sier}^0(r)$. In the first one with very small values of $\log(r)$, from $-8.3$ to $-5.5$, $E_0$ is about $-0.361$; this behavior is due to the fact that, having very small cells and being the dataset finite, the number of cells intersected by the geometries tends to be rather constant (one polygon for each cell), but having polygons instead of points, it slightly decreases. (ii) In the second interval, from $-4.1$ to $-0.7$, the behavior of the fractal emerges and $E_0$ is about $-1.578$, namely very close to the expected theoretical value. For the plot $BC_{sier}^2(r)$ similar considerations can be done, since, as theoretically proved, fractals have equal exponents for every parameter $q$ considered in the power law.*

*For the dataset $D_{PRaus}$, again two intervals are detected, in the first one $E_0$ is around $-1.2$. This means that the dataset has a skewed distribution (indeed, $E_0 > -2$), and the more we reduce the size of $r$, the more emerges the local behavior of the dataset as a line: indeed, it is composed of linestrings. In the second interval $E_0$ is around $-1.554$, which means that for greater values of $r$ the dataset is present in almost all cells, so the diffusion is close to the uniform one and thus $E_0$ is close to $-2.0$. In such cases $E_2$ has to be considered, since it is able to capture the real distribution of the dataset, which is actually different from its diffusion. In particular, for $D_{PRaus}$ we can observe that most of the roads are concentrated in the south east and south west of Australia.*

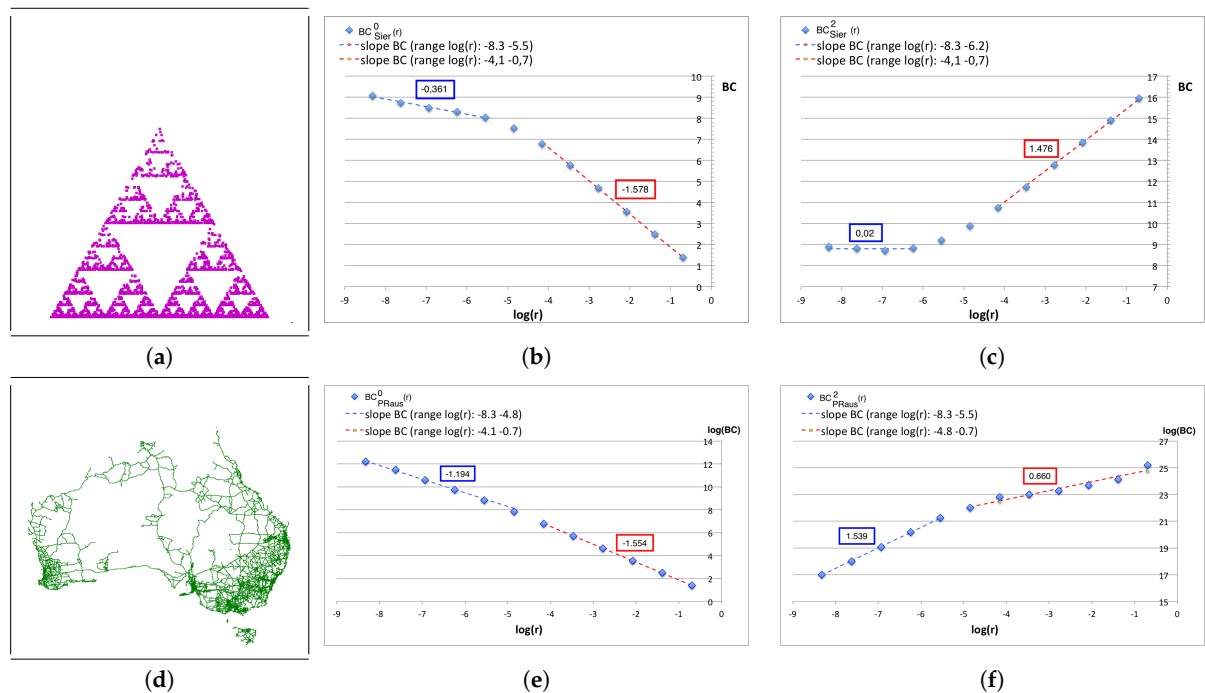

**Figure 1.** Example of Box-counting plot for: (**a**–**c**) a synthetic dataset with the distribution of a Sierpinski's triangle, and (**d**–**f**) a real-world dataset representing the primary roads of Australia.

In order to practically use $E_0$ and $E_2$ as indicators of distribution for real datasets, it is necessary to find an easy and efficient way for computing them given a dataset $D$. The next section presents a MapReduce implementation in SpatialHadoop of an algorithm for efficiently computing $E_0$ and $E_2$.

## 4. A MapReduce Algorithm for $E_0$ and $E_2$

This section presents the second contribution of the paper, namely a MapReduce algorithm for the computation of the exponents $E_0$ and $E_2$ of the power laws introduced in Equation (2). It was implemented w.r.t. SpatialHadoop, thus some basic spatial functions are assumed to be available in the target system.

Given a dataset $D$ containing geometries of different types, the required exponents can be obtained by first computing the box-counting function $BC_D^q(r)$ for different values of $r$, and then by using linear regression to determine the slope of the line representing the plot of $BC_D^q(r)$, as shown in Figure 1. Such slope is equal to the parameter $E_q$ which we need to estimate. It follows that the main goal is the computation of the Box-counting plot of $D$ for $BC_D^0(r)$ and $BC_D^2(r)$; the successive linear regression can be applied in constant time, since the computed plots will always have the same number of pairs $(log(r), log(BC_D^0(r))$ (or $(log(r), log(BC_D^2(r)))$).

In order to compute the required Box-counting plots, we need to know the reference space of $D$, which is represented by its MBR. The user can already know this MBR, thus passing it as a parameter, or it can be unknown. In the latter case, a preliminary MapReduce job can be performed to compute it, for instance by invoking the method FILEMBR() of SpatialHadoop. Given the MBR of $D$, it is now necessary to choose a list of grids $(G_1, \ldots, G_n)$ with increasing cell size $r_1, \ldots, r_n$, to be used for computing the series $(r_1, BC_D^0(r_1), \ldots)$ and $(r_1, BC_D^2(r_1), \ldots)$. The most convenient choice for the grids is the one that can guarantee to produce Box-counting plots with homogeneous distributed values in a logarithmic scale. Thus, starting from a value $r_0$, at each step $i$ we derive the $r_i$ values by multiplying $r_{i-1}$ by 2. The initial $r_0$

is chosen so that $r_n$ is less than the side of the MBR (to simplify the presentation we suppose here that the MBR is a square) and that a minimum number of grids could be generated.

In the MapReduce procedure, each map task will process one split of $D$, and for each geometry $g$ of the split it identifies the cells of the finer grid $grid_0$ that it intersects. it counts the the number of geometries intersected by each cell of $grid_0$ and at the end, it writes as result a set of pairs $\langle i, p_i(D) \rangle$, where $i$ identifies the cell of $grid_0$, while $p_i(D)$ is the count of geometries in $D$ intersecting the cell $i$. Notice that we do not raise such count to the power $q$ at this point because the value $p_i(D)$ is not final as it needs to be merged with the result of other map tasks during a subsequent reduce phase.

The map task is presented in Algorithm 1. The finer grid $grid_0$ is generated in the SETUP() method (line 4). Additionally, a hash map $bcount_0$ is created (line 5) that will store the counts of (only) the not empty cells of $grid_0$. In this way we avoid to store all the cells of $grid_0$. In the MAP() method (lines 6–9), for each input geometry *geom*, the function $grid_0.intersects(geom)$ is executed (line 7). This function returns a list of identifiers representing the cells of $grid_0$ intersected by *geom*. Finally, for each one of such cells $c_j$, the hash map $bcount_0$ is updated (line 9). In the CLEANUP() method (lines 10–12), the content of $bcount_0$ is written to disk producing the input of the successive reduce task.

---

**Algorithm 1:** Map Task

---

1  **class** MAPPER
2    $grid_0, bcount_0$
3    **method** SETUP($r_0, MBR$)
4      $grid_0 \longleftarrow Grid(r_0, MBR)$
5      $bcount_0 \longleftarrow HashMap()$

6    **method** MAP(*id, geom*)
7      $intCells \longleftarrow grid_0.intersects(geom)$
8      **foreach** $c_j \in intCells$ **do**
9        $bcount_0.put(c_j, bcount_0.get(c_j) + 1)$

10   **method** CLEANUP()
11     **foreach** $\langle c_j, p_k \rangle \in bcount_0$ **do**
12       WRITE($c_j, p_k$)

---

The complexity of each map task is $O(N_{split})$, where $N_{split}$ is the average number of geometries of $D$ per split.

The pseudo-code of the reduce task is presented in Algorithm 2. Notice that its SETUP() method generates a list of grids *grids* and the corresponding list of hash tables *bcounts*. The grids are generated starting from an initial cell side $r_0$, while successive $r_i$ values are obtained by multiplying $r_{i-1}$ by 2.

---

**Algorithm 2:** Reduce Task

---

1   **class** REDUCER
2    *grids, bcounts*
3    **method** SETUP($r_0$, *MBR*)
4     *grids, bcounts* $\longleftarrow \varnothing$
5     $r \longleftarrow r_0$
6     **while** $r < MBR.width$ **do**
7      *grids.add*($Grid(r, MBR)$)
8      *bcounts.add*($HashMap()$)
9      $r \longleftarrow r * 2$

10    **method** REDUCE($c_j$, *counts*)
11     *total* $\longleftarrow 0$
12     **foreach** $v \in counts$ **do**
13      *total* $\longleftarrow total + v$
14     **foreach** $i \in [0.. \mid grids \mid]$ **do**
15      **if** $i = 0$ **then**
16       $bc_0.put(c_j, total)$
17      **else**
18       $gr_i \longleftarrow grids.get(i)$
19       $c_j \longleftarrow gr_i.getCell(c_j)$
20       $bc_i.put(c_j, bc_i.get(c_j) + total)$

21    **method** CLEANUP()
22     $E_0, E_1 \longleftarrow \varnothing$
23     **foreach** $g_i \in grids$ **do**
24      $bc_i \longleftarrow bcounts.get(g_i)$
25      $BC^0 \longleftarrow bc_i.size()$
26      $BC^2 \longleftarrow 0$
27      **foreach** $\langle c_j, p_k \rangle \in bc_i$ **do**
28       $BC^2 \longleftarrow BC^2 + p_k^2$
29      $E_0.put(\log(g_i.cellSize()), \log(BC^0))$
30      $E_2.put(\log(g_i.cellSize()), \log(BC^2))$
31     WRITE($0, regressionSlope(E_0)$)
32     WRITE($2, regressionSlope(E_2)$)

---

The REDUCE() method receives as input the partial results produced by the map tasks. In particular, for each cell $c_j$ of $grid_0$ it receives a list of partial counts for $c_j$ computed by the various map tasks. Therefore, it initially sums such counts producing its total count (line 13). These total values are stored into the hash map $bc_0$ (lines 16). The same total values are used for updating the cells of the other grids that contain $c_j$ (lines 20), so that every hash map $bc_i$ can be filled during the same iteration.

In the CLEANUP() method the series $(r_1, BC_D^0(r_1), \dots)$ and $(r_1, BC_D^2(r_1), \dots)$ are computed starting from the hash maps $bc_i$ (lines 23–30). Finally, the slopes representing $E_0$ and $E_2$ are computed applying the procedure *regressionSlope* (lines 31–32). In this phase the range of scales can be split in some intervals when necessary in order to obtain a constant slope.

The complexity of each reduce task is $O(n_{cell} + n_{grid})$, where $n_{cell}$ represents the total number of cells of $grid_0$ intersected by the dataset $D$, while $n_{grid} =\mid grids \mid$.

Table 3 illustrates the results of the application of the proposed MapReduce job to both synthetic and real datasets. The first column describes the dataset in terms of its distribution and number of geometries (i.e., cardinality), while the second column reports the taken time in milliseconds and the last column contains the obtained values.

**Table 3.** Algorithm application for the computation of $E_0$ and $E_2$.

| Dataset (Cardinality) | Total Time (ms) | $E_0/E_2$ |
|---|---|---|
| Sierpinski's triangle (5,000) | 1115 | $-1.578/1.522$ |
| Sierpinski's triangle (500,000) | 3187 | $-1.584/1.520$ |
| Uniform distribution (4,500,000) | 27,721 | $-2.000/1.992$ |
| Diagonal (4,500,000) | 17,244 | $-1.131/1.068$ |
| Double cluster (4,500,000) | 28,311 | $0.000/0.178$ |
| Primary Roads USA (12,400) | 2982 | $-1.273/1.230$ |

## 5. Heuristics for Choosing the Best Partitioning Technique

Given the MapReduce procedure illustrated in the previous section, it is now time to introduce some criteria for choosing the right partitioning technique based on the computed values. In particular, we need to introduce some criteria assessing the quality of the partitioning techniques with respect to the operations that we want to perform onto the indexed datasets. Considering the spatial join, range query and $k$-nn, we define two quality descriptors that have to be minimized in order to improve the effectiveness of an index on a dataset $D$.

- $d_1(D)$: the %RDS (relative standard deviation with respect to the mean) of the split cardinality (i.e., the number of geometries);
- $d_2(D)$: the percentage of the reference space covered by the grid that represents dead space, that is, space containing no data.

Notice that, $d_1$ affects the cost of a single map task, while $d_2$ has an impact on the total number of map tasks to be instantiated by the desired operation.

Let us consider again the partitions produces by applying the techniques illustrated in Section 2 on synthetic and real datasets, and in particular the grids obtained in Table 2. For each of them Table 3 reports the computed values of $E_0$ and $E_2$. Such obtained results confirm the ability of $E_0$ and $E_2$ in distinguishing cases that need different partitioning techniques. Indeed, for the uniform distribution ($E_0 \sim -2.0$ and $E_2 \sim 2.0$) the obtained partitions are very similar for all techniques; in this case the regular grid can be the best choice, since its creation cost is less. For the diagonal with buffer ($E_0 \sim -1.0$ and $E_2 \sim 1.0$) the Quadtree-based and the Rtree-based grids adapt best to the dataset distribution. However, the partitioning produced by the Rtree-based grid has more balanced cells w.r.t. the Quadtree-based partition, in terms of number of geometries per cells. Finally, when a clustered dataset is considered ($E_0 \sim 0.0$ and $E_2 \sim 0.0$), we obtain the best partitioning with the Quadtree-based grid, while the Rtree-based grid produces a partition with lots of dead space.

At this point the idea is to exploit $E_0$ and $E_2$ in order to choose the right grid for data partitioning, without building all kinds of indexes. This goal can be obtained thanks to the following properties of $E_0$ and $E_2$.

**Property 1** (Dataset diffusion). *Given a dataset D, when its exponent $E_0$ is close to $-2.0$, then the descriptor $d_2$ for any index is close to zero, that is, no dead space exists.*

**Proof.** If $E_0$ is equal to $-2.0$, then the dataset is distributed in the reference space in such a way that, for each scale $r$ of the grid $G$ that is considered during the box-counting computations (see Definition 1), all the cells of $G$ are intersected by the geometries of $D$. Considering a reference space $1 \times 1$:

$$E_0 = -2 \implies BC_D^0(r) = r^{-2}, thus$$

$$\sum_i p_i(D)^0 = \frac{1}{r^2} = count(all \ cells \ of \ the \ grid)$$

Therefore, the geometries of $D$ are spread throughout the reference space and no dead space exists. □

**Property 2** (Dataset distribution). *Given a dataset D, when its exponent $E_2$ is close to 2.0, then the descriptor $d_1$ of a regular grid is close to zero, that is, every cell of the grid contains the same number of geometries belonging to D.*

**Proof.** If $E_2$ is equal to 2.0 then the dataset is distributed in the reference space so that the box-counting computations (see Definition 1) produce the following result:

$$E_2 = 2 \implies BC_D^2(r) = r^2, thus$$
$$\sum_i p_i(D)^2 = \alpha r^2$$

Since the term $\sum_i p_i(D)^2$ is minimized when the $p_i(D)$ are all equal, then we obtain the uniform distribution of the dataset, which is our thesis. □

Now we add to the above presented formal properties some experimental observations. Given a dataset $D$:

1. When the computed $E_0$ is around 1.0, then the dataset is skewed, has some dead space and is located around a curve, thus it is usually connected. In this case, the regular grid will have high values for the descriptors $d_1$ and $d_2$, since the dataset cannot be uniformly partitioned into cells with equal area, while the Rtree-based grid will have the lowest value for $d_2$, because it is the technique that starts from geometries and not from the space for clustering data, but also a good value for $d_1$, since data cover a connected region. Finally, the Quadtree-based grid will have a good value for $d_1$, but a higher value for $d_2$ w.r.t. Rtree-based grid.
2. When the computed $E_0$ is around 0.0, then the dataset is skewed, has lots of dead space and is located around two or more points, thus it is usually not connected. In this case, the regular grid will have high values of $d_1$ and $d_2$, as before; the Rtree-based grid will have the lowest value for $d_2$ but a higher value for $d_1$ due to the fact that the connectivity is lost, while the Quadtree-based grid will have a better values for both $d_1$ and $d_2$ than the Rtree-based one, since it adapts better to the clustered datasets.
3. Similar considerations are valid for the values of $E_2$, where instead of dead space it detects regions of lower/higher concentration, thus affecting more deeply the descriptor $d_2$.

Using Properties 1 and 2 and the above listed observations, we propose the following heuristic for choosing the best partitioning technique customized to a dataset $D$. It is based on the decision tree shown in Figure 2.

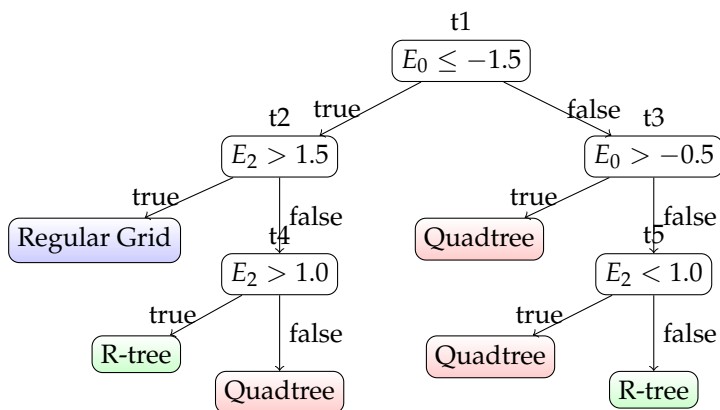

**Figure 2.** Decision tree for the choice of the more appropriate partitioning technique for a dataset $D$.

The first test $t_1$ on $E_0$ determines if the dataset spread throughout the reference space or it is concentrated in some way. If $t_1$ is true, we need to verify the kind of distribution it has using $E_2$. In particular, if $t_2$ is true, then the dataset has a uniform distribution and the best index is the regular grid. Conversely, if $t_2$ is false, we need an additional test $t_4$ on $E_2$ to check if the occupied space is connected or not. If $t_4$ is true, then the dataset can be considered connected and the Rtree-based grid is chosen; otherwise, the Quadtree-based grid will be the best choice.

On the contrary, if $t_1$ is false, then the dataset has a skewed distribution in the reference space. In this case the test $t_3$ on $E_0$ is applied to determine if the dataset is clusterized in some way. In particular, if $t_3$ is true, then the Quadtree-based grid will be chosen; conversely, the final test $t_5$ on $E_2$ is used to determine some degree of connectivity and to choose between a Quadtree or an Rtree index.

Notice that in the tree we use threshold values equal to $-1.5$, $-0.5$ for the choices regarding $E_0$, while we use threshold values equal to 1.0, 1.5 for $E_2$, since we consider $E_2$ only when the dataset is spread throughout the reference space or when $E_0$ is around 1.0, thus in this case the values near to 0.0 cannot be reached by $E_2$. The effectiveness of the proposed heuristic has been tested by meas of some experiments illustrated in the next section.

## 6. Experiments

This section presents some experiments performed on a Hadoop cluster composed of 10 slave nodes and 1 master node, on which Hadoop 2.8 and SpatialHadoop 2.4 extension have been installed. Each node is characterized by 4 cores, 8GB of RAM memory and 1TB of SSD, all nodes are connected through an infinity band network. The experiments consider a collection of datasets containing both synthetic and real spatial data. Table 4 illustrates their characteristics.

**Table 4.** g used in the experiments: UD = uniform distribution, DL = diagonal line, $DL_B$ = diagonal line with buffer, $DC_B$ = double cluster with buffer, $PR_{USA}$ = USA primary roads, $PR_{AUS}$ = Australian primary roads, $WA_{USA}$ = USA water areas and $ST_{AUS}$ = Australian states. In the 6th column: RT = Rtree, QT = Quadtree, and RG = regular grid.

| | Dataset | Size | $E_0$ | $E_2$ | Index |
|---|---|---|---|---|---|
| Synthetic | **UD** | 1.27 GB | −2.000 | 1.997 | RG |
| | **DL** | 1.07 GB | −1.131 | 1.067 | RT |
| | **$DL_B$** | 1.07 GB | −1.762 | 1.177 | RT |
| | **$DC_B$** | 1.07 GB | −1.615 | 0.042 | QT |
| Real | **$PR_{USA}$** | 1.00 GB | −1.273 | 1.032 | RT |
| | **$WA_{USA}$** | 2.17 GB | −1.837 | 1.602 | RG |
| | **$PR_{AUS}$** | 1.18 GB | −1.554 | 0.660 | QT |
| | **$ST_{AUS}$** | 0.32 GB | −1.731 | 1.692 | RG |

For each dataset, we computed $E_0$ and $E_2$ (columns 4 and 5 of the table) by applying the algorithm presented in Section 4 and we determined the best index (column 6) according to the decision tree presented in the previous section. we considered three kinds of operations on such datasets: range query, spatial joinan $k$-nn, using all the three analyzed indexes, that is, we applied three partitioning techniques: regular grid (RG), Quadtree-based grid (QT) and Rtree-based grid (RT). As regards to the range query operation, we performed 200 range queries for all datasets and for all kind of indexes by varying the side length of the query region (from 0.001 to 0.05 with respect to a reference space normalized to $1 \times 1$). The obtained results are shown in Table 5 where the reported times are an average value over about 200 queries for each case. Notice that, the best performances are always obtained when the used index is the one suggested by the proposed heuristic. In particular, when the dataset is uniformly distributed (**UD**) all techniques have similar performances, indeed the difference between the best and the worst solution is at most 5%. For the other synthetic datasets, we obtained as expected that for both datasets having a distribution around a line, the best is the Rtree-based grid, with a higher gain in the case of the **DL** dataset. Instead, when we observed a higher similarity with a clustered distribution, like for the **$DC_B$** dataset, the best solution becomes the Quadtree-based grid. Real datasets shown the same behavior of the synthetic datasets confirming the quality of the proposed heuristic. In particular, when the dataset is nearly uniformly distributed, as for **$WA_{USA}$** or **$ST_{USA}$** the differences induced by the various partitioning techniques are very low.

For the spatial join operations, we performed some experiments on synthetic datasets by joining a uniform distributed dataset, called **UN**, with another dataset $D$ having one of the other kind of considered distributions. In particular, for each kind of distribution in Table 4, 10 synthetic datasets are randomly generated and joined with **UN**. The obtained results are reported in Table 6. As for the previous tests, except for the uniformly distributed case where all indexes have the same performances, the best results are obtained when $D$ is partitioned using the suggested kind of index, achieving also a considerable reduction on the execution times.

For real datasets we performed a first join between datasets **$PR_{USA}$** and **$WA_{USA}$** and a second join between **$PR_{AUS}$** and **$ST_{AUS}$**. Both joins were performed in 9 different conditions, that are obtained by considering all the possible combinations of index type on the first and second dataset. The results are reported in Tables 7 and 8. The best execution times is obtained when both datasets are partitioned with the suggested kind of index. Notice that, the worst performances are obtained when a regular grid is built for both datasets, confirming that it is not a good index in case of skewed data.

**Table 5.** Result of experiments with range queries. we performed experiments on each listed dataset, considering queries of size 0.001, 0.005, 0.01 and 0.05 w.r.t. a normalized reference space of $1 \times 1$. The reported time is in milliseconds and is an average value over about 200 queries for each case.

| Data | QA | Range Query with Index | | | % Improv. |
|------|------|------|------|------|------|
| | | RG | QT | RT | bst vs wst |
| **UD** | 0.001 | 16,730 | 16,341 | 16,567 | 1% |
| | 0.005 | 16,986 | 16,368 | 16,823 | 4% |
| | 0.010 | 16,915 | 16,362 | 16,194 | 4% |
| | 0.050 | 21,913 | 20,872 | 21,266 | 5% |
| **DL** | 0.001 | 14,689 | 11,818 | **11,773** | 20% |
| | 0.005 | 14,317 | 11,561 | **11,497** | 20% |
| | 0.010 | 12,581 | 9597 | **9530** | 22% |
| | 0.050 | 14,650 | 11,483 | **11,204** | 22% |
| **DL$_B$** | 0.001 | 15,187 | 12,890 | **12,803** | 16% |
| | 0.005 | 14,052 | 11,230 | **10,471** | 25% |
| | 0.010 | 13,139 | 11,570 | **11,022** | 16% |
| | 0.050 | 13,358 | 11,467 | **11,288** | 16% |
| **DC$_B$** | 0.001 | 12,798 | **10,886** | 13,253 | 18% |
| | 0.005 | 13,754 | **12,501** | 15,454 | 19% |
| | 0.010 | 13,729 | **12,319** | 15,431 | 20% |
| | 0.050 | 13,129 | **12,296** | 15,854 | 22% |
| **PR$_{USA}$** | – | 13,532 | 13,756 | **12,102** | 12% |
| **WA$_{USA}$** | – | **11,593** | 11,711 | 11,854 | 2% |
| **PR$_{AUS}$** | – | 21,343 | **18,398** | 19,396 | 14% |
| **ST$_{AUS}$** | – | **19,274** | 19,929 | 19,396 | 3% |

**Table 6.** Result of experiments with joins on synthetic datasets. The **UN** has been indexed with a regular grid, while $D$ is partitioned with various kind of index. The reported time is in milliseconds and is the average of the times of about 10 queries for each kind of dataset distribution for $D$.

| UN ⋈ $D$ | | | Index on $D$ | | | |
|------|------|------|------|------|------|------|
| | RG | | QT | | RT | |
| $D$ | #Maps | Time | #Maps | Time | #Maps | Time |
| **UD** | 37 | 145,307 | 51 | 150,458 | 54 | 147,646 |
| **DL** | 35 | 125,327 | 60 | 96,001 | 22 | **40,205** |
| **DL$_B$** | 35 | 123,920 | 67 | 123,920 | 27 | **50,412** |
| **DC$_B$** | 19 | 103,410 | 27 | **48,083** | 27 | 61,776 |

**Table 7.** Result of experiments with joins on real datasets regarding the primary roads and the water areas of the USA.

| | | | Index PR$_{USA}$ | | | |
|------|------|------|------|------|------|------|
| Index | RG | | QT | | RT | |
| WA$_{USA}$ | #Maps | Time | #Maps | Time | #Maps | Time |
| RG | 124 | 547,848 | 89 | 333,084 | 74 | **316,355** |
| QT | 228 | 438,277 | 96 | 342,126 | 87 | 332,858 |
| RT | 126 | 443,466 | 79 | 352,038 | 65 | 345,754 |

**Table 8.** Result of experiments with joins on real datasets regarding the primary roads and the states of Australia.

| Index on $ST_{aus}$ | Index $PR_{AUS}$ | | | | | |
|---|---|---|---|---|---|---|
| | RG | | QT | | RT | |
| | #Maps | Time | #Maps | Time | #Maps | Time |
| RG | 25 | 253,957 | 34 | **126,168** | 24 | 180,687 |
| QT | 34 | 166,135 | 27 | 161,827 | 31 | 177,809 |
| RT | 24 | 171,213 | 31 | 172,511 | 25 | 167,005 |

Finally, for the *k*-nn operation, we perform some experiments on both synthetic and real data. As regards to the synthetic ones, we produce for each kind of considered distributions 10 different datasets and for each of them we perform 40 experiments by randomly choosing the query point, the first half of experiments considers a value of *k* equal to 100, while the second one a value of *k* equal to 10,000. Similar experiments have also been performed for real datasets, by varying both the value of *k* and the reference point. The results are reported in Table 9. As for the previous operations, in case of uniform distributed datasets, the difference between the best and the worst indexing technique is very low. Conversely, in the other cases the choice of the right partitioning technique has a significant impact on the overall performances.

**Table 9.** Result of experiments with *k*-nn queries. we performed experiments on each listed dataset, considering *k* equal to 100 and 10,000. The reported time is in milliseconds and is an average value over about 20 queries for each case.

| Data | *k* | *k*-nn Query with Index | | | % Improv. |
|---|---|---|---|---|---|
| | | RG | QT | RT | bst vs wst |
| **UD** | 100 | 30,405 | 31,536 | 31,461 | 4% |
| | 10,000 | 33,692 | 33,705 | 34,471 | 2% |
| **DL** | 100 | 61,228 | 57,080 | **52,916** | 14% |
| | 10,000 | 61,289 | 58,522 | **52,322** | 15% |
| **$DL_B$** | 100 | 75,238 | 55,053 | **54,602** | 27% |
| | 10,00 | 68,917 | 65,563 | **56,980** | 17% |
| **$DC_B$** | 100 | 66,725 | **62,138** | 71,743 | 13% |
| | 10,000 | 71,240 | **65,818** | 66,019 | 8% |
| **$PR_{USA}$** | 100 | 51,647 | 74,473 | **50,223** | 33% |
| | 10,000 | 79,608 | 69,239 | **54,735** | 31% |
| **$WA_{USA}$** | 100 | **36,039** | 41,989 | 51,471 | 30% |
| | 10,000 | **39,052** | 52,158 | 73,942 | 47% |
| **$PR_{AUS}$** | 100 | 45,354 | **36,086** | 82,493 | 56% |
| | 10,000 | 63,900 | **49,212** | 113,476 | 57% |

## 7. Conclusions

This paper considers the impact of a skewed distribution on the performances of three spatial operations, that is, range query, spatial join and *k*-nn, with particular attention on its effect on the balancing of the work performed by the map tasks during their parallel execution. we considered as reference framework SpatialHadoop and its partitioning techniques: regular grid, Quadtree-based grid and R-tree based grid. Such partitioning techniques can produce different results on the basis of the distribution exposed by each dataset, and these results can lead to potentially great differences in the performances of spatial operations. Therefore, the choice of the right partitioning technique for each kind of dataset becomes

an important activity in order to completely exploit the benefit of a MapReduce execution. For this reason, we proposed a new technique based on the Box-counting function [14] for efficiently estimating a dataset distribution and accordingly choose the more suitable partitioning technique. Some experiments on synthetic and real datasets have been performed to show the effectiveness of the proposed heuristic. Such experiments reveal that in all cases the suggested partitioning technique is able to improve the execution time of the following spatial operations. In particular, while in presence of uniform distribution, all partitioning techniques have essentially the same effect on the following operations, when skewed datasets are considered, the choice of the wrong partitioning technique can double the time required for a given analysis. Moreover, when the distribution resemble a linear concentration, data-based paritioning techniques (like the R-tree) are more suitable, since they are able to produce more balanced splits, while they are less suitable for clustered datasets because they could produce partitions with lots of dead spaces. In such case, space-based partitioning (like the Quadtree) are the right choice.

Future work regards the application of the knowledge about dataset distribution to reduce-side joins, the extension of the proposed approach to selectivity estimation, the application of the box-counting function for estimating the skewness of multidimensional datasets, also outside the spatial context. Another important extension could be application of the proposed method in order to define new mixed partitioning techniques. More specifically, the technique could be applied to break a given datasets into homogeneous subsets on which different partitioning technique could be applied. It will also be worthwhile in future work to explore the direction that decides the partitioning without heuristics, based purely on the identified distribution and by means of machine learning models.

**Author Contributions:** Conceptualization, Alberto Belussi, Sara Migliorini and Ahmed Eldawy; Formal analysis, Alberto Belussi, Sara Migliorini and Ahmed Eldawy; Methodology, Alberto Belussi, Sara Migliorini and Ahmed Eldawy; Writing—original draft, Alberto Belussi, Sara Migliorini and Ahmed Eldawy. All authors have read and agreed to the published version of the manuscript.

**Funding:** This work was partially supported by the Italian National Group for Scientific Computation (GNCS-INDAM) and by "Progetto di Eccellenza" of the Computer Science Department, University of Verona, Italy.

**Conflicts of Interest:** The authors declare no conflict of interest.

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
