# Peer review of "Skewness-Based Partitioning in SpatialHadoop"

_ijgi, doi:10.3390/ijgi9040201_

Round 1

Reviewer 1 Report

This is thorough work choosing systematically the best partitioning technique for spatial data. It provides analytical foundations, heuristic methods, and demonstration with suitable data sets.

It will be interesting to see if the method can be used to break datasets into subsets for which different partitioning methods are near optimal. It will also be worthwhile in future work to explore doing the partitioning without heuristics, based purely on the identified distribution.

Reviewer 2 Report

The manuscript shows most of the same ideas and concepts already presented in a previous work of the authors ( reference 16) . In many sections even the wording is quite similar to reference 16. Table 1, 3 and Figure 1 have been taken from reference 16, too. It looks like there is little additional research in the new work, but more detailed explanation of the algorithms. In other words, new contribution is scarce, just more details are been added to the previous work. The manuscript should not be published in the present form.

Reviewer 3 Report

The paper describe a way to easily detect some hints about the dataset distribution and based on these choose the more effective partitioning technique to apply. The topic is interesting and worth of publishing. However, my main point concerns the very structure of the text.

There is no clearly separated methodological part. It can be concluded that chapters 2, 3 and partly 4 belong to it. But in present form it is difficult to select what is Authors new contribution, and what is existing knowledge. I would recommend to reorganize this part of the paper to underline what is Authors personal input.

The same with Results. In Experiments section there is some methodological assumptions together with the results. In my opinion assumptions should be moved to the Methodology, and results should be developed.

Conclusion should be developed. It sounds rather like summary than discussion. There is a lack of critical reference to the results and explanation of the usefulness of proposed solutions.

There is no relation to Figure 1 in the text (before figure). The same with Table 2.

Relation to Alg.2 is after algorithm, should be before. The same with Figure 2.

The same with others. Please check the rest of figures/tables etc. Relation in the text should be always before the figure/table.

Reviewer 4 Report

A technique based on both a box counting function and a heuristic is proposed for detecting the degree of skewness of an input spatial dataset and then deciding which partitioning technique to apply in order to improve as much as possible the performance of subsequent operations.

Although the experiments on both synthetic and real datasets are presented to support the effectiveness of the proposed approach, I believe this fine work can be completed by having/adding an analysis and comparison with other existing techniques.

Authors then may add a table with metrics/parameters that shows their proposed technique benchmarks the current techniques.

Round 2

Reviewer 4 Report

I still believe that the authors need to compare their work with other related techniques. 

It doesn’t have to be the techniques based on both box counting function and heuristic, but can be the techniques based on either of them.